# One Health: How Interdependence Enriches Veterinary Ethics Education

**DOI:** 10.3390/ani10010013

**Published:** 2019-12-19

**Authors:** Joachim Nieuwland, Franck L. B. Meijboom

**Affiliations:** Faculty of Veterinary Medicine, Utrecht University, 3584 CM Yalelaan 2, The Netherlands; f.l.b.meijboom@uu.nl

**Keywords:** one health, veterinary medicine, veterinary ethics, veterinary education interdependence, pedagogy, moral agency, mindfulness

## Abstract

**Simple Summary:**

The idea of One Health acknowledges the interdependence of human and non-human animal health against the backdrop of a shared environment. This requires collaboration across disciplines to tackle complex health problems. How does One Health affect the education of veterinary ethics, traditionally restricted to both animal and professional ethics? First, veterinary ethics education provides an opportunity within the curriculum for students to engage with the meaning and implication of One Health, so as to develop their own viewpoint. Similarly, One Health can enrich veterinary ethics. It does so by introducing relevant ethical fields and other cultural perspectives, as well as promoting ways of teaching that motivate and sensitize students to become aware of the underlying interdependency and complexity of health issues while at the same time fostering their capacity for ethical problem solving.

**Abstract:**

What does One Health imply for veterinary ethics education? In order to answer this question, we will first have to establish what One Health itself involves. The meaning and scope of One Health, however, cannot be established without reference to its values—whose health matters? Veterinary ethics education is well equipped to facilitate such an open-ended inquiry into multispecies health. One Health also widens the scope of veterinary ethics by making salient, among other fields, environmental ethics, global health justice, and non-Western approaches to ethics. Finally, One Health requires students to engage with interdependence. Discussing three levels of interdependence, we argue that veterinary ethics stands to benefit from a more contemplative pedagogy.

## 1. Introduction

Since the inception of the discipline, societal demands have had an impact on veterinary education. The first Faculty of Veterinary Medicine was established in Lyon, France, in 1761 predominantly to address the devastating consequences of *Rinderpest morbillivirus* [1]. Non-human animal (hereafter “animal”) diseases and their impact continue to shape veterinary curricula until this day, while societal expectations have spurred the development of professional ethics. In recent decades, veterinary education has become increasingly attuned to both veterinary ethics and One Health.

Veterinary ethics is a relatively young and specific field within the wider range of applied moral philosophy, falling in line with increased philosophical interest in animals in the second half of the 20th century [2,3]. Bernard Rollin [4] and Jerold Tannenbaum [5] are recognized as pioneers of veterinary ethics, with an important difference between them, namely the status of animal ethics. Rollin employed the broad range of animal ethics including his own work on animal rights, while Tannenbaum restricted veterinary ethics to the bounds of veterinary care specifically. Notwithstanding their differences, both have contributed greatly to making veterinary ethics an integral part of the curriculum at many Faculties of Veterinary Medicine around the globe [6].

An upsurge in emerging infectious diseases at the beginning of the 21st century provided the main impetus in the development of One Health. The Manhattan principles, a list of 12 recommendations made by health experts in 2004 to address the rising threat of emerging infectious diseases, clearly pointed towards a more holistic approach to health policy: “Recent outbreaks of West Nile Virus, Ebola Hemorrhagic Fever, SARS, Monkeypox, Mad Cow Disease and Avian Influenza remind us that human and animal health are intimately connected. A broader understanding of health and disease demands a unity of approach achievable only through a consilience of human, domestic animal and wildlife health—One Health” [7].

Veterinarians have embraced One Health early on, with the publication of the American Veterinary Medical Association’s (AVMA) report *One Health: A New Professional Imperative* leaving no doubt about its endorsement [8]. The AVMA defines One Health as “the collaborative efforts of multiple disciplines working locally, nationally, and globally, to attain optimal health for people, animals, and our environment.” [8].

Until now, veterinary ethics and One Health have remained largely isolated from each other within veterinary education. Only recently have philosophers begun to explore the ethics of One Health (see Section 3 for a more comprehensive overview). We are interested not only in the interplay between (veterinary) ethics and One Health but also, and most importantly, its implications for veterinary ethics education: What new ethical issues (should) arise for veterinary students by taking this multispecies perspective on health? How should veterinary ethics education engage with One Health so as to make it relevant to the development of future veterinary professionals?

## 2. Veterinary Ethics Education: Facilitating Critical Engagement with One Health

While the above mentioned AVMA definition is often referred to when explaining the idea of One Health, it remains rather abstract and open to interpretation. For some, One Health is largely about zoonotic diseases and antimicrobial resistance, while for others, it also includes benefits to human health that accrue from animal use such as animal assisted therapy and companionship [9]. As some have pointed out, where to draw the line is certainly not straightforward: “Interrogation of the concept of One Health is overwhelming in its scope: it is challenging to think of examples of issues that might not be classified as One Health challenges” [10] (p. 53). An apt illustration of this challenge is the recent development of One Welfare, drawing upon One Health but extending it even further by bringing out the interconnections between humans, animals, and the environment not only in terms of health but also with regard to welfare [11,12]. How do we deal with such a plethora of One Health perspectives, especially in light of developing veterinary curricula?

One Health, moreover, unmistakably involves value assumptions and we should acknowledge that [13,14]. In addition to creating a better overview of relevant drivers and determinants of health, One Health involves a call for collaboration across disciplinary lines to “attain optimal health for people, animals, and the environment” [8]. What “optimal health” involves, and how to allocate our resources and moral consideration across these domains in order to attain it, inevitably becomes a matter of values and moral judgment [13,14].

Veterinary ethics education can play a vital role in exploring the idea of One Health and its moral implications, especially as it has already done so with regard to other significant concepts of veterinary medicine within the curriculum. For instance, veterinary ethics education generally makes students reflect on the meaning and implications of animal welfare. Equipped with the ability to engage in critical and self-reflective deliberation, as well as conceptual analysis of such key concepts for the veterinary profession such as animal welfare [15,16], veterinary ethics education is more than suited for the job of making sense of One Health. We can point out to veterinary students the normative dimension of One Health, as well as encourage to explore these for themselves.

Does such questioning destabilize One Health? Instead, opening up One Health to critical engagement will likely do quite the opposite. To follow the analogy with animal welfare, ongoing debate has not led to the demise of animal welfare. Rather, animal welfare has attracted miscellaneous perspectives and interpretations, challenging both students and teachers to think for themselves when it comes to such concepts overtly susceptible to interpretation. In order to make sure that everyone involved understands and actively endorses the definition used for, for example, animal welfare, one has to critically engage with such concepts. Otherwise, Babylonian confusion and selective appropriation may raise their unwelcome heads. While the debate on animal welfare has not been settled, some core assumptions are shared by many despite their differences. Similarly, as we have argued elsewhere, many descriptive characteristics of One Health, for instance, interspecies disease interfaces as well as cross-species transfer of health knowledge, can be supported by an overlapping consensus composed of many different perspectives [17]. Such a shared understanding can function as a starting point for One Health policy, as well as an invitation to explore other interpretations. Rather than fixating One Health for the purpose of aligning veterinary education, we propose to see veterinary ethics education as the facilitator of One Health conversations, as part of an open-ended inquiry into multispecies health and veterinary responsibility.

## 3. One Health: Ways for Veterinary Ethics Education to Branch Out

Following endorsement of One Health in education and health policy, philosophers have started exploring the implications of One Health ethics. Given the emphasis put on the interconnections between the three domains (human, animal, and the environment), it is not surprising to imagine novel perspectives arising out each of these domains, as well as in relation to the others. As we see it, One Health challenges us to further develop Rollin’s vision of veterinary ethics education as an amalgam of professional ethics and animal ethics in the following five ways:

### 3.1. The Ecological Turn in Veterinary Ethics

First, in addition to the traditional subjects, animal ethics and professional ethics that have defined veterinary ethics since its beginning, environmental ethics appears most relevant to address ecological concerns that interrelate with human and animal health concerns [18,19]. Environmental ethics has until now not significantly contributed to veterinary ethics, which has been largely dominated by individualist animal ethics, in part because of the emphasis on patients and professional duties in a clinical context. Albeit not necessarily absent, discussion of environmental ethics perhaps remained something of a more theoretical exercise, to contrast individualist animal ethics instead of providing a coherent and systematic outlook relevant to professional concerns of veterinary practice. Moreover, to the extent that veterinary medicine does relate to the collective level, for instance populations in livestock farming, typical environmental ethical concerns including biodiversity loss, wildlife habitat degradation, and climate change have not fully entered mainstream veterinary ethics. With the recognition of the interdependence between the health of humans and animals against the background of their shared environments, veterinary ethics will have to take an ecological turn.

### 3.2. Progress in Animal Ethics

Second, adding an ecological dimension to human and animal health also brings up new questions from an animal ethics perspective, further developing the ethical discipline considered by many central to veterinary ethics. For example, zoonotic disease surveillance, monitoring, and control raises questions about the extent to which risks can be shared across species [20], squarely placing wild animals within the ambit of veterinary ethics. Should we reevaluate the acceptability of culling both in livestock-production as well as wildlife-management [14,21]? A multispecies approach to health knowledge, for example by means of biobanks [22] or epidemiology of obesity [23] raises questions about distribution of resources, the way in which these are institutionalized and how they translate into health benefits. These questions are not so much ethical as they are political, dovetailing with the recent political turn in animal ethics [24,25,26], another opportunity for veterinary ethics to branch out and develop.

### 3.3. Health Justice

Third, One Health also challenges veterinary students to engage with human health policy. Here, we find ourselves confronted with concerns of public health ethics and global health justice: What level of risk imposed by zoonotic diseases is acceptable from a public health perspective? To what extent should international health policy be tailored to health needs across geographical areas? How should the international community respond in case of an infectious disease outbreak? What does a sustainable, just, and healthy food production require?

### 3.4. Multidisciplinary, Interdisciplinary and Transdisciplinary

As already apparent in some of the examples given, the three domains in ethics overflow into each other. How do these fields relate to each other in the context of veterinary ethics? Here, a distinction between multi-, inter-, and transdisciplinary approaches proves helpful. Including other ethical perspectives is at first a multidisciplinary move; it “draws on knowledge from different disciplines but stays within the boundaries of those fields” [27] (p. 359). Sometimes, multidisciplinary evolves into a more in-depth and structural collaboration between disciplines. For instance, Rollin’s multidisciplinary idea of using both professional and animal ethics has culminated in a whole new field of applied ethics in the veterinary curriculum. Veterinary ethics displays all the marks of being interdisciplinary, as it “analyzes, synthesizes and harmonizes links between disciplines into a coordinated and coherent whole” [27] (p. 359). Similarly, adding new ethical fields to the arsenal of veterinary ethics, as One Health invites us to do, will most likely also involve a shift from multidisciplinary to interdisciplinary. Moreover, as One Health emphasizes interdependence, an opportunity opens up for a transdisciplinary approach, which “integrates the natural, social and health sciences in a humanities context, and in so doing transcends each of their traditional boundaries” [27] (p. 359). So rather than culminating in a new discipline—as the route from multidisciplinary to interdisciplinary goes—transdisciplinary approaches avert recoil within disciplinary boundaries, instead looking for ways to continuously communicate across such boundaries. Teaching veterinary ethics checks all the boxes for fostering such transdisciplinary integration, as it has to navigate in-between moral theory and the demands of the future working environment of veterinary students—both of which characterized by multi- and interdisciplinary collaboration.

### 3.5. Non-Western Approaches to Ethics

One Health involves not just as shift towards multidisciplinary, interdisciplinary and transdisciplinary approaches in veterinary ethics. One Health also, by opening up a more global perspective on health policy, prompts a critical reflection of the historical roots underlying veterinary ethics. Doing so uncovers a strong emphasis on Western Philosophy, with Singer, Regan, and Rollin as respectively, albeit remotely, the exponents of Benthamian, Kantian, and Aristotelian thinking. If we truly take up the challenge of One Health of working across disciplinary divides, and do so in the light of globalization, we ought not to remain within the lines drawn by cultural tradition that has shaped veterinary ethics ever since its beginning. What can veterinary ethics education learn from African, Asian, South American, and a multitude of indigenous ethical views, and how should they be incorporated in global perspectives on multispecies health policy?

## 4. Interdependence: Enriching Veterinary Ethics Education

One Health affirms the interdependence between humans, animals, and ecological processes. The need to respond to emerging infectious diseases required an acknowledgment that human health is interdependent with the natural world at large. Zoonotic diseases, with the majority finding their source in wildlife [28], have pressed upon health professionals the recognition that human health is vulnerable to disease outbreaks at the wildlife-interface [7]. Moreover, anthropogenic impact on the environment represents a significant driver in the emergence of such diseases, making humans both vulnerable as well as implicated in the rise of emerging zoonoses [29]. Of course, veterinary medicine has been engaged with the interdependence of human health and zoonotic diseases before the rise of One Health, with Veterinary Public Health as a prime example. Nonetheless, One Health arguably takes interdependence beyond the confines of Veterinary Public Health, ushering somewhat of a paradigm shift in health policy [10,30].

While definitions of One Health often include the term interdependence, its precise meaning generally remains unspecified or conflated with the term interconnectedness [31]. Such ambiguity is unfortunate, especially given the potentially novel implications for veterinary ethics education of taking interdependence seriously. A dictionary entry of interdependence explains it as “the fact of depending on each other” [32]. Much hinges, as we will discuss below, on how one interprets this rather general definition of interdependence.

### 4.1. Interdependence as Interface

A first way of looking at interdependence is to broaden one’s initial scope, to include up- and downstream factors that influence health objectives. For veterinarians, this means, for example, not to only focus on domesticated animals but to be keenly aware of the interface these animals share with wildlife, and the possible ramifications of these relations in terms of infectious disease. Interdependence entails being aware of disease-interfaces. Such a view brings animals forth in our moral deliberations that may have been underappreciated before, something not necessarily to their benefit. One Health may even entail a net harm to animals when they are identified primarily in terms of sources of disease and threat to human health [33]. Thus, the widening of scope does bring up new moral questions but does not necessarily imply an expanding circle of moral consideration. Interdependence is viewed largely in descriptive terms, not carrying much normative weight itself. The moral values that have shaped health policy until then remain vested, applied within a more ecological outlook. If indeed the scope is only widened incrementally, with human interests as both the starting point for ecological inquiry as well as the evaluative delimitation, perhaps it is better to understand this in terms of “intradependence” rather than interdependence. Those in favor of keeping One Health free from One Welfare appear at the side of intradependence. They might see disease-interfaces as the defining feature of One Health thinking. Those arguing for a merger of the two, push interdependence somewhat further, seeing One Health also in terms of health-interfaces; a way of seeing that naturally recognizing the links between health and wellbeing.

### 4.2. Individual Interdependence

Another interpretation of interdependence goes beyond the incremental broadening of the scope of disease ecology by reconsidering the individual. Whereas becoming more aware of interfaces already to some extent embeds humans and domesticated animals into a myriad of ecological processes, this second approach not only embeds but also appreciates humans and animals as ecosystems themselves, as multispecies collectives. Recent scientific insights, for example with regard to the human (and animal) microbiome, put pressure on the idea of understanding ourselves as independent individuals in a biological sense [34]. Instead, humans and animals form an interdependent multispecies collective together with the manifold organisms that populate their bodies, something that potentially changes what we consider appropriate health interventions.

As Jonathan Beever and Nicolae Morar put it, 

if microbes are an *interdependent* (not just a merely interconnected) part of who we are, then any attempt to unwillingly alter our microbiome might well be considered a personal assault rather than a mere modification of our environmental conditions. Changing the ways we think about how things are related (biologically) invites us to reconsider our value conceptions and the value priorities that orient us, ethically, to problems of health and the methods by which those values are enacted.[31] (p. 186; emphasis original)

We need such examples in order to distinguish between interdependence and interconnection, both of which are used generally interchangeably in the One Health literature, a conceptual ambiguity that Beever and Morar take issue with. Why? The relation between the individual and their microbiome is exemplary of interdependence. While ecological processes evidently connect humans and animals in many ways, we need not view all sorts of relations as interdependence; sometimes interconnection suffices. While interconnection involves a relation of some sort, interdependence involves something more, they argue, “a mode of relationality that is crucial for the health of an organism and for delimiting some important moral boundaries between units of care” [31] (p. 193). The microbiome not only changes the way we see individuals, it also should, they argue, change the way we treat them. This differentiates the interdependence found in individual animals from the way in which they are bound up in socio-ecological relations. As they point out, “we know that not everything matters the same or is influential to our health to the same degree. For example, while climatic conditions do have an impact on one’s health, the intensity of their input into one’s health condition is not on a par with the metabolic input that human organisms receive from their microbiome” [31] (p. 192)

Still, one could argue that interdependence calls for an even deeper appreciation in veterinary ethics education of what it means to be ecologically embedded. In other words, in the end, the second view of interdependence remains within the ambit of intradependence. Understanding an animals’ microbiome as intricately part of who they are involves a reevaluation of the individual that stays beneath the skin. It conflates interdependence with local symbiosis between organisms. Paradoxically, it may end up reaffirming the individual as something separate from ecological processes. From an ecological point of view, such a localized understanding of interdependence is not without its problems, as it removes from view the myriad ecological processes that are necessary for organisms to exist, let alone flourish, in the first place. Interdependence perhaps requires us to go beyond the epidermal layers that separate animals from their “environment”. For instance, humans will not survive when the oxygen level in the atmosphere dives considerably below its current percentage of 20.95% [35]. When temperature exceeds 35 °C for extended periods of time, humans perish due to heat stress [36]. That these ecological conditions are generally stable does not necessarily make them any less worthy in comparison to embodied physiological processes. A dysfunctional ecosystem could wreak havoc just as much as a microbiome in disarray. The separation between internal physiological and external ecological processes affecting one’s health reflects an individualization unwarranted given the way the health of humans and animals fundamentally depends on both of these conditions. Moreover, it is exactly this ecological and interspecies web of relations that One Health uncovers and highlights, which is why a deeper sense of interdependence would enrich veterinary ethics education.

### 4.3. Interbeing

We can take interdependence further, by toppling any delimitation of interdependence; it is interdependence all the way. Thich Nhat Hanh captures this thorough sense of interdependence with the word interbeing: “If you are a poet, you will see clearly that there is a cloud floating in this sheet of paper. Without a cloud, there can be no rain; without rain, the trees cannot grow; and without trees, we cannot make paper. The cloud is essential for the paper to exist. If the cloud is not here, the sheet of paper cannot be here either. So, we can say that the cloud and the paper *inter-are*” [37] (p. 27; emphasis original). Nhat Hanh sets no limits, in contrast to the first view on interdependence, as he goes on to include the sun, the logger, the wheat of the logger’s daily bread, etc. in his ever-expanding illustration of interbeing. “Everything—time, space, the earth, the rain, the minerals in the soil, the sunshine, the cloud, the river, the heat, and even consciousness—is in that sheet of paper” [37] (p. 28). In agreement with the second view, Nhat Hanh points out that humans “are made of non-human elements” [37] (p. 74) but sees no reason to restrict interdependence to symbiotic relations such as those between individuals and their microbiome: “To be is to inter-be with every other thing” [37] (p. 28). Beyond reconceptualizing individuals as multispecies collectives, which in itself already offsets new normative perspectives on protecting and promoting individual health, understanding individuals as thoroughly interdependent both internally and externally could bolster a normative commitment of promoting collective health.

How are we to understand such a shift towards a more collective view of health? Some worry that such thoroughgoing interdependence translates directly in a sort of ecological egalitarianism, in line with the tenets of Deep Ecology as envisioned by Arne Næss [38]. On that view, the worry goes, interdependence represents a box of Pandora, undercutting not only anthropocentrism of traditional health policy, but also breaching any newly installed boundaries as a rationale to put human health objectives above those of non-humans [31]. Such egalitarianism mutes any decision-making in health policy, because “if everything matters, then it is difficult if not impossible to discriminate appropriately between value claims” [31] (p. 192).

If nothing exists independently, however, we need not necessarily value each constituent of any relation equally in moral terms. As part of Deep Ecology, Næss indeed argued for biospherical egalitarianism: “*the equal right to live and blossom* is an intuitively clear and obvious value axiom” [38] (p. 96; emphasis original). Such a moral outlook is not necessarily implied by understanding interdependence as interbeing. While part of the novelty of interdependence for veterinary ethics involves more engagement with environmental ethics, interbeing is not limited to an environmental outlook, or biocentric perspective, only. It does not prescribe the moral equality of all species and their members in the process. It allows for a sentiocentric moral orientation pace a biocentric one [39]. For instance, seeing oneself as interdependent does not make everything that makes you an individual a recipient of moral concern. Suffering and happiness, already key concepts for veterinary ethics, can continue to provide moral guidance, now against the backdrop of interbeing.

Nonetheless, with interdependence as interbeing, the purview of veterinary moral agency is potentially widened extensively, as a delineation of moral concern becomes equivocal. Furthermore, interbeing also highlights the students’ own interdependence, for example, by uncovering the extent to which cultural-historical backstories shape their individual moral agency, or how their moral decision-making plays out in a working-environment subject to a range of stakeholder demands. Notwithstanding the heightened moral sensibility interbeing creates, in order to align this increased complexity of moral decision making with veterinary ethics education, we have to take a pedagogical turn.

## 5. Pedagogy of Veterinary Ethics Education

In what way do the above described perspectives translate into veterinary education? To begin with veterinary ethics education’s role of scrutinizing One Health, we have already suggested a similar approach to the conceptual and ethical analysis of animal welfare. The aim is to foster a reflective attitude to such concepts not only to stimulate critical reflective capacities of students themselves, but also to help develop One Health in veterinary curricula, to make it more robust by opening it up to deliberation regarding its ambit.

Perhaps more difficult is the extent to which we should diversify and make multidisciplinary the veterinary ethics curriculum itself. Should every veterinary student, for example, ponder questions about distributive justice with regard to international pharmaceutical development? Although this will remain an open-ended inquiry in part, certain criteria are at least relevant. A good starting point is the general academic competency veterinary students should acquire with regard to such broader ethical questions. One Health provides an opportunity to bring in ethical debates prominent in society but only indirectly implicated with veterinary medicine, for instance ecological concerns around climate change and biodiversity loss. Being able to engage intellectually with these broad issues can be seen as an unmistakable competency of academics, even more so if one’s profession is, also if largely indirectly, implicated.

In addition to a general academic competency, we suggest a track-specific competency. The further students’ progress throughout the study, the more specific questions of veterinary ethics education should become by making use of relevant disciplines in various ways. The general competency provides a basis for deepening students’ awareness of and ability to address One Health prompted ethical considerations. It builds upon the veterinary responsibility for public health. If veterinarians already bear responsibility for public health by way of preventing and controlling the outbreak of zoonotic infectious disease [6], it does not appear far-fetched to attune veterinary moral agency to the upstream (for example, deforestation for the sake of growing crops destined to feed domesticated animals; use of animal research for pharmaceutical or food production) and downstream (emission of CO_2_, CH_4_, etc.; impact of free-roaming cats on bird and small mammal populations) ecological considerations of diverse human-animal interrelations. These examples provide some more detail for an enriched track-specific veterinary ethics education.

At what point to delineate veterinary responsibility remains a question, but in any case, it extends beyond the traditional setting of veterinary ethics, where Rollin likens veterinarians to either pediatricians or garage-mechanics [4]. Not only do veterinarians have to balance between these positions with regard to the moral status of the patient and the associated obligations for themselves as health professionals, they also have to determine the extent to which they pay attention to moral issues intersecting with their everyday clinical practice. It is up to us, teachers of veterinary ethics, among others, to triage on these matters.

We have to be careful, however, to foster veterinary moral agency rather than undermine it by confronting students with potentially overwhelming complexity. Especially the third interpretation of interdependence—interbeing—might end up paralyzing veterinary students. Moreover, moral problems in veterinary practice already represent a great challenge, not to mention the perils of moral distress that arise when professionals are insufficiently able to live up to their own moral ideals because of the way in which their working environment and its demands structure and restrict individual agency [40].

How do you teach veterinary student to be aware of interdependence whilst fostering their ability to make sound ethical judgments? Assuming One Health does raise the challenges for veterinary ethics as indicated, we have to look deeper into the pedagogy, the how of teaching ethics in the veterinary curriculum. Often, students are presented with the major theories of animal ethics, using this theoretical background to deliberate and deal with ethical issues in the clinical context [4]. The emphasis on animal ethics as well as clinical cases appears ill-suited to foster awareness of interdependence that One Health brings forth. Moreover, the pedagogical style of traditional veterinary ethics education emphasizes, through the use of animal ethics theory, reasoning and individual autonomous agency. Dealing with interdependence, we suggest, calls for a more contemplative pedagogical style [41].

A contemplative pedagogy aims to create awareness. It can do so in various ways, including, but also going beyond, ethical reasoning. For example, one could ask students to bring a recipe, or even invite them to bake, an ethically approved cake. Students taking our elective course of veterinary ethics (MA level, non-track specific, Faculty of Veterinary Medicine, Utrecht University) come with various examples: using self-grown products only, local or national, cheap, fair trade, vegan, organic, without the use of plastic materials, with the lowest emission of greenhouse gasses, family recipe, cultural tradition, etc. The point is not necessarily to establish the ethically approved cake, rather to raise students’ awareness of interdependence apparent in their everyday lives, as well as to try to understand where others are coming from.

Another contemplative approach (also part of the above-mentioned elective course in veterinary ethics) involves embodied encounters with animals in a context atypical given the cultural status associated with this species. Animal farm sanctuaries provide such an opportunity. Rather than merely discussing animal welfare in reference to theories of animal ethics in a classroom, students can engage with the animals and deliberate, in a non-judgmental fashion, on what makes an animal’s life good. Experiencing animals in such a non-standard environment can highlight the extent to which one’s values are culturally shaped in a way that theoretical informed reasoning is unable to provide. Interdependence also applies to one’s own moral agency.

Of course, veterinarians have to operate in a force-field of various responsibilities, obligations, demands and expectations, of which One Health appears only to make more complicated. However, letting complexity and conflicting demands narrow and blunt moral agency should be prevented if only to protect the wellbeing of veterinarians themselves. Paradoxically, numbing oneself in order to deal with moral distress can easily backfire [40], which is why becoming aware of the ways in which one’s agency is shaped over time, embedded within a certain environment, and confronted with a myriad of diverging expectations, could help to bolster robust moral agency.

Interdependence, by way of a contemplative pedagogy, sheds light on the role of veterinary ethics education in supporting future veterinary professionals in ethical decision-making but also safeguard individual wellbeing in doing so. The issue of moral distress in veterinary practice should make us reflect on the way in which veterinary ethics education can help to diminish its detrimental effects on wellbeing, as well as foster veterinary moral agency. Here, mindfulness meditation, a key element of a contemplative pedagogy, could align student and professional wellbeing with veterinary ethics education. Mindfulness meditation and its benefits to alleviate stress among veterinary students have been explored in several studies [42,43,44]. In addition to lowering stress, mindfulness can also play a vital role in cultivating moral agency. It does so by making individuals aware of the way in which they perceive the world around them [45]. In part, this complements the emphasis on critical reflection already present in traditional veterinary ethics, for instance, when students are confronted with a clinical case and challenged to provide their judgment and supporting argumentation. However, it also counterbalances the need for reasoning, opening up one’s perception to salient moral features of a situation. By doing so, it downplays the likelihood of individuals blunting their moral perception by ways of rationalization. At the same time, heightening moral perception addresses the problem of an ever-increasing complexity and interdependence promulgated by One Health, difficult to overcome by rational deliberation. While One Health calls for multi-, inter- and transdisciplinarity at the level of veterinary ethics, these perspectives should function within veterinary ethics education as a means to develop, as much as possible, perceptive and skillful moral action. Along these lines, mindfulness meditation provides an opportunity to further accommodate veterinary ethics education to the demands of veterinary practice, especially against the backdrop of interdependence introduced by One Health.

## 6. Conclusions

One Health provides a relevant framework for health professionals, including veterinarians, to cut through the complexity arising from interdependence between humans, animals, and ecosystems. However, the way one understands this perspective on health and its implications involves value assumptions, which is why, from an ethical perspective, students need to critically engage with One Health. Veterinary ethics education is well placed and equipped to support students in doing so. Like many all-encompassing concepts (such as sustainability and animal welfare), One Health is vulnerable to selective interpretation, Babylonian confusion and appropriation. Using its familiar strengths of fostering critical reflection and conceptual analysis, veterinary ethics education should encourage students to scrutinize One Health and develop their own perspective.

One Health also enriches veterinary ethics education as we know it. The call for collaboration across disciplinary boundaries urges veterinary ethics to branch out beyond the traditional pair of professional ethics and animal ethics, with opportunities for multidisciplinary, interdisciplinary and even transdisciplinary approaches. Broadening veterinary ethics is not limited to including relevant fields such as environmental ethics only, as One Health also facilitates multicultural exchange of perspectives. Finally, by championing the idea of interdependence, One Health incites veterinary ethics education to take a pedagogical turn.

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
