# Peer review of "One Health: How Interdependence Enriches Veterinary Ethics Education"

_animals, 2019, doi:10.3390/ani10010013_

Round 1

Reviewer 1 Report

Overall, this paper is original, clearly written, and potentially contributes to the literature. It aims to inform and modify veterinary ethics education by relying on One Health. It also discusses the ethics and general implications of the OH approach, with its calls to interdependence etc.. I thus think it should be accepted for publications. I do include however suggestions for minor revision in the text- none should be required for publication.

Author Response

Dear reviewer,

Thank you very much for your comments and feedback. Please find our point-by-point response to the feedback annotated in the text. 

Best regards,

Franck Meijboom and Joachim Nieuwland

Reviewer 2 Report

Reviewer comments for manuscript ID animals-667711 entitled ‘One Health: how interdependence enriches veterinary ethics education’

General comments

A well written and thoroughly researched manuscript on nascent and emerging fields of veterinary ethics education and One health. I congratulate the authors for explaining the philosophical and moral aspects of the subject. I would like to specially mention the excellent write up on environmental and ecological ethics. The discussion on environmental ethics is quite novel, interesting and thought provoking. Ecological implications of animal ethics in managing domestic and wild life population management provides future perspectives to One health.

I have some concerns on Section 4 - Let’s take a broader perspective: interdependence – suffers with lack of clarity, repetition and at times confuses the reader. I suggest the authors to please have relook of this section to make it more cohesive and linked to the main theme. This has been the only weak link of the manuscript.

Conclusions should have a take home message for the readers – the way forward and future implications. Though it has been discussed but needs more robust interpretation.

Specific comments

Simple summary

Line 12: Replace ‘nonhuman’ with ‘non-human’ and maintain this consistency all through the manuscript.

Line 19: Replace ‘learn ……..’ with ‘ motivates and sensitizes ………’

Abstract

Line 22-24: I do not understand this sentence. Please clarify. ‘Providing a specific interpretation, however, reflects one’s values, which is why discussing the import of One Health in itself already enriches veterinary ethics’

Introduction

Line 41: Delete ‘a’ from ‘……. a professional ethics’

Line 44-47: Reframe the sentence ‘Veterinary ethics is a relatively young and specific field within the wider range of applied moral philosophy, ensuing philosophical interest in nonhuman animals of, most notably, Peter Singer [2] and Tom Regan [3]. Bernard Rollin [4] and Jerold Tannenbaum [5] are recognized as pioneers of veterinary ethics, with an important difference between them, namely the status of animal ethics.’ As ‘Veterinary ethics is a relatively young and specific field within the wider range of applied moral philosophy, ensuing philosophical interest in non-human animals. Peter Singer [2], Tom Regan [3], Bernard Rollin [4] and Jerold Tannenbaum [5] are recognized as pioneers of veterinary ethics as a discipline, with an important difference between them, being the status of animal ethics.’

Line 48: Delete ‘Whereas’

Line 49: Insert ‘while’ just before ‘… Tannenbaum’

Line 51: Replace ‘unmistakable’ with ‘integral’

Let’s talk about One Health

Line 92: Please elaborate what values in the sentence ‘One Health unmistakably involves values and we should acknowledge that’

Lines 94-100: Please simplify these lines. The explanations are contradictory to the descriptions given in previous sentences.

Lines 102-105: Again here also the sentences contradict each other. There is no clear message.

Let’s work together

Line 131: Replace ‘…. we see…’ with ‘… imagine.’

Lines 165-66: Please delete ‘… as much international as national,’

Line 171: I suggest ‘aim at’ is a better word than ‘look like’?

Line 186: Replace ‘…each of their’ with ‘… beyond their..’

Line 188-89: Replace ‘…such a return to a disciplinary boundary…’ with ‘… recoil within disciplinary boundaries’

Line 191-92: Please delete ‘…. – itself, as we have seen, already largely an interdisciplinary practice ‘

Line 193-97: This is a huge sentence. Please reword it , to make it more clear. I suggest ‘Veterinary Ethics education has to foster an attitudinal shift towards the implications of One Health in veterinary curricula. It has a potential to drift away from siloed thoughts of humanities and medical science on multispecies health policy towards interdependence’.

Let’s take a broader perspective: interdependence

Line 218: Please delete ‘…. thoroughly interdependent’.

Line 223-26: Please delete this paragraph. It does not provide any novel information. ‘What exactly is interdependence? While often used in One Health literature, its precise meaning often remains unspecified [30]. More generally, it is defined as “the fact of depending on each other” [31]. Dependence cuts both ways. Much hinges, as we will see, on how one interprets this rather general definition of interdependence’.

Lines 282-83: Please explain this sentence more clearly. I am not able to follow this point. ‘Still, one could argue that interdependence calls for an even deeper appreciation of the ways in which individuals are shot through ecologically’

Lines 285-294: Please explain these lines more lucidly for the reader. These is an important information and needs more clarity for better comprehension of the reader ‘

Line 319: Please delete ‘renders’ and pluralize ‘mute’

Conclusions

Line 443: Replace ‘making sense of’ with ‘understanding and imbibing’

Line 453: Replace ‘… something of a shift in pedagogy’ ‘with a pedagogical shift’

Author Response

Dear reviewer,

Thank you very much for you comments and feedback. Please find attached  the annotated manuscript with modifications made as well as a point-by-point response to two more general questions below.

Best regards,

Franck Meijboom and Joachim Nieuwland

Response to Reviewer 2 Comments

Point 1: I have some concerns on Section 4 - Let’s take a broader perspective: interdependence – suffers with lack of clarity, repetition and at times confuses the reader. I suggest the authors to please have relook of this section to make it more cohesive and linked to the main theme. This has been the only weak link of the manuscript.

Response 1: We have more clearly introduced the discussion about what interdependence means within One Health as an introduction to the three interpretations we discuss ourselves. With regard to the second interpretation, we have both clarified it more extensively as well as discuss why we think there is a need for an even deeper appreciation of interdependence in One Health. This latter point ties it together with the discussion of the third interpretation, which we have supplemented with a brief discussion of implications for veterinary ethics education.

By emphasizing the discussion about interdependence within One Health as apparent in the literature, as well as connecting all three interpretation with each other, as well as the final section on pedagogy, we think that the “enriching” character of interdependence becomes more salient.

Point 2: Conclusions should have a take home message for the readers – the way forward and future implications. Though it has been discussed but needs more robust interpretation.

Response 2: we have rewritten part of the conclusion so as to show our steps in the paper, as well as emphasized the concrete actions this requires of those working in the field of veterinary ethics education (critically engage with One Health, explore beyond animal ethics and professional ethics, and invest in a more contemplative pedagogy).
